# Finding the Best Programmable PWM Pattern for Three-Level Active Front-Ends at 18-Pulse Connection

**Alexander S. Maklakov** *, **Tao Jing**, **Andrey A. Radionov, Vadim R. Gasiyarov and Tatyana A. Lisovskaya**

Department of Mechatronics and Automation, South Ural State University, 454080 Chelyabinsk, Russia; jingtao19940214@gmail.com (T.J.); radionov.mail@gmail.com (A.A.R.); gasiyarovvr@gmail.com (V.R.G.); lisovskaiata@susu.ru (T.A.L.)
* Correspondence: alexandr.maklakov.ru@ieee.org; Tel.: +7-(351)-2723230

**Abstract:** The existing publications on the analysis of power quality indicators in modern electric power supply systems are void of a comprehensive approach to improving these indicators in power systems by implementing multipulse connections. To the authors' knowledge, this paper is the first to analyze current harmonic distortions in an 18-pulse connection of three-level active front-ends (AFE) featuring a programmed PWM. Raw data were obtained from, and current quality was analyzed for the power circuit of the main electric drive actuating the rolls in the rolling stand of a plate mill. The key feature of such circuitry is that the synchronous motor of each work roll is connected to the grid with an 18-pulse connection that uses three phase-shift transformers, where the phase shifts are $0°$ (delta/delta), $20°$ (delta/polygon) and $−20°$ (delta/polygon). The circuitry connects three frequency converters (FC) with the AFEs in parallel. Phase-shift transformers were found to periodically overheat in the process. When overheating occurred, a programmed PWM voltage waveform was applied where harmonics 17 and 19 were eliminated. The goal and objectives were to analyze why the transformer would overheat and to find out how the issue could be addressed. The authors developed a simulation model of the research object in order to assess power quality parameters. Simulation results obtained in Matlab/Simulink were used to estimate the total harmonic distortions (THD) and individual harmonic factors for up to the 50th secondary transformer winding and grid harmonic with four different programmed AFE PWM voltage waveforms. The results helped find the best such waveform to prevent phase-shift transformers from overheating; one with harmonics 5, 7, 17 and 19 eliminated. The experimental and mathematical modeling results in the paper were confirmed by positive effects after industrial implementation of the system. Research performed directly on the operating equipment has been classified by the company and is not publicly available. These results are highly versatile and could be used in similar research on other circuitries to ensure the electromagnetic compatibility of nonlinear power-consuming devices.

**Keywords:** power converters; 18-pulse connection; power quality; active front-end; programmed PWM; selective harmonic elimination



## 1. Introduction

Major industrial facilities nowadays commonly use multipulse connection to the grid to power up regenerative medium-voltage (MV) adjustable speed drives (ASD) [1]. Multipulse connections are a simple and effective way to reduce the effects of semiconductor power converters on the quality of grid voltage [2]. Compared to connections that use six pulses, multipulse connections have two core advantages:

- lower total harmonic distortion (THD) [3];
- lower individual harmonic factors for some voltage and current harmonics [4].

In turn, frequency converters (FCs) in the regenerative MV drives contain three-level active front-ends (AFE) and three-level inverters that are capable of:

- recovering electricity to the grid when braking;

- maintaining zero shift of the fundamental current harmonic with respect to the input voltage;
- compensating for the reactive power at the grid connection point;
- using programmed PWMs in order to comply with the low and medium-frequency voltage and current quality standards [5].

Despite being common and having been researched intensively from electromagnetic compatibility requirements, AFEs remain an issue in terms of their effect on the quality of grid voltage. Literature reviews suggests the following technical solutions are relevant today [6,7]:

- using multipulse connection to the grid based on multiwinding phase-shift transformers [8];
- use of programmed PWM voltage waveforms to eliminate or mitigate selected harmonics, i.e., Selective Harmonic Elimination PWM and Selective Harmonic Mitigation PWM [9];
- use of passive L and LCL filters to filter out higher harmonics on the AFE AC side [10];
- connecting MV regenerative ASD to a separate substation.

## 2. Statement of Problem, Goals and Objectives

A literature review helped formulate the key problem covered herein; which pertains to the development of an optimal PWM algorithm for high-power AFEs based on the NPC topology. Research of indicators contributing to the output voltage and current harmonics of AFE is associated with studying PWM control of AFEs. However, the existing publications on the analysis of power quality indicators in modern electric power supply systems are void of a comprehensive approach to improving these indicators in power systems by implementing multipulse connections.

Industrial use of the existing regenerative drives based on multipulse connections shows that stepdown phase-shift transformers tend to overheat. This issue has been reported for the 18-pulse main drive of the rolling stand of a plate mill. Preliminary analysis indicated the issue was mainly attributable to the AFEs. When overheating occurred, a programmed PWM voltage waveform was applied where harmonics 17 and 19 were eliminated. Overview of the earlier tested 18-pulse AFE-based grid connections revealed lack of an in-depth analysis into the issue.

This gave rise to the goal of this research, i.e., to analyze harmonic distortions of currents in an 18-pulse connection of three-level AFEs featuring various programmed PWM AC voltage waveforms. Raw data were obtained from, and current quality was analyzed for, the power circuit of the main electric drive actuating the rolls in the rolling stand of a plate mill; it therefore served as the object of this research. To that end, the research team developed a simulation model of this object using transfer functions and structural modeling in Matlab/Simulink. The analysis produced the best recommendable programmed PWM voltage waveform for AFEs to prevent phase-shift transformers from overheating.

The paper is structured as follows: Section 1 for introduction and relevance; Section 2 for statement of problem, goals, and objectives; Section 3 for description of the object; Section 4 for description of the control system; Section 5 for simulation results and discussion; Section 6 for conclusions.

## 3. Object of Research

### 3.1. Specifications of the Object

Research was carried out on the main electric drive of the rolling stand of a plate mill. Figure 1 shows the key components of its grid connection. Table 1 shows the motor specifications. Each motor is controlled by three FCs connected in parallel to three-level AFEs, see Figure 1. Table 2 shows the AFE specifications.

**Table 1.** Motor specifications.

| $U_r$, V | $I_r$, A | $f_r$, Hz | $P_r$, MW | $\cos(\varphi)$ | $R_l$, mOhm | $L_l$, mH |
|---|---|---|---|---|---|---|
| 3300 | 2460 | 10 | 12 | 1 | 9.54 | 32.15 |

**Table 2.** Three-level AFE specifications.

| $U_r$, V | $I_r$, A | $U_{dcr}$, V | $f_{sw}$, Hz | $P_r$, MW | Efficiency$_r$, % | $C_{dc}$, μF |
|---|---|---|---|---|---|---|
| 3300 | 800 | 5020 | 350 | 8.4 | 97 | 6341.54 |

The key feature of such circuitry shown in Figure 1 is that FCs are connected to the grid by an 18-pulse connection that uses three phase-shift transformers where the phase shifts are 0° (delta/delta), +20° (delta/polygon) и −20° (delta/polygon). The transformers have identical parameters, see Table 3.

**Table 3.** Parameters of phase-shift transformers: 0°, +20° and −20°.

| $S_r$, kVAR | $U_{1r}$, V | $U_{2r}$, V | $I_{1r}$, A | $I_{2r}$, A | $U_{sc}$, % | $\Delta P_{sc}$, kW | $\Delta P_{nl}$, kW |
|---|---|---|---|---|---|---|---|
| 5700 | 10,000 | 3300 | 329.1 | 997.2 | 16 | 55 | 4.9 |

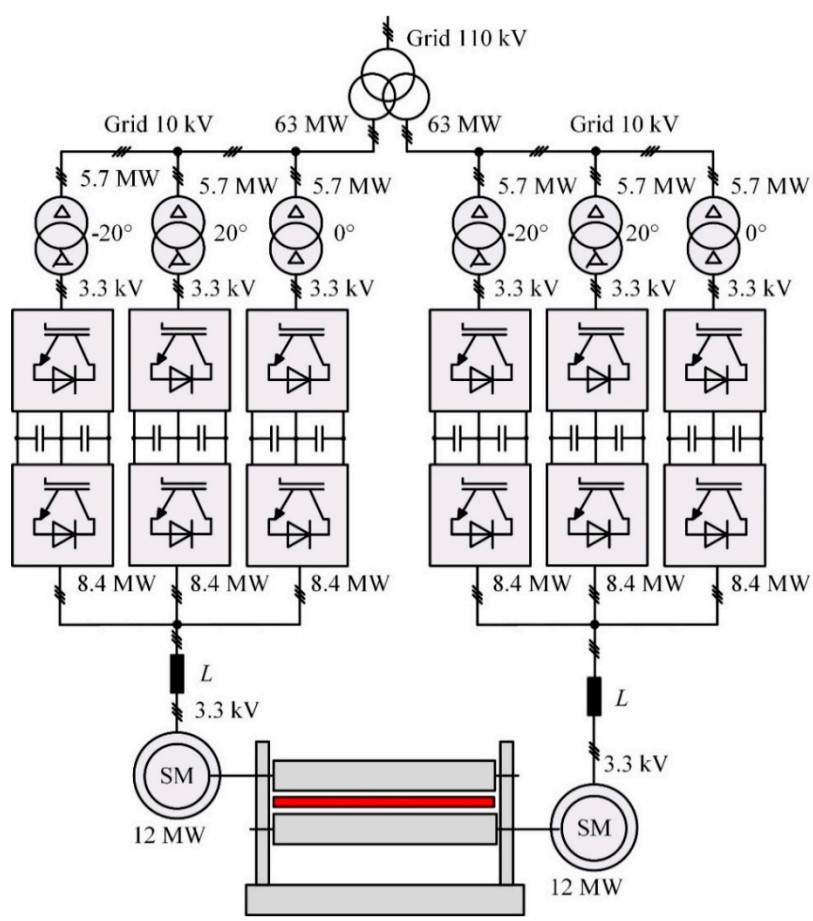

**Figure 1.** Object connection 18-pulse circuit diagram.

## 3.2. 18-Pulse Connection

Multipulse technology is a solution that provides greater total power and addresses the electromagnetic compatibility issues of nonlinear power-consuming equipment. The

18-pulse connection of the analyzed object only has the $18n \pm 1$ (n = 1, 2, ..., ∞) significant current harmonics. This is attained by shifting the first harmonics of secondary voltages in phase-shift transformers by 0° and ±20° [11].

A +20° shift in primary voltage vectors against secondary voltage vectors in first harmonics is attained by splitting the primary winding into two sections and connecting these windings in such a way that the magnetic fluxes of this phase and of the adjacent phase are counter-directed, as shown in Figure 2a. Such connection causes the vectors to sum up and make a +20° shift against the voltage vector of the greater section. Ratios between the primary winding parts are calculated using primary voltage vector diagrams, see Figure 2b,c,e,f).

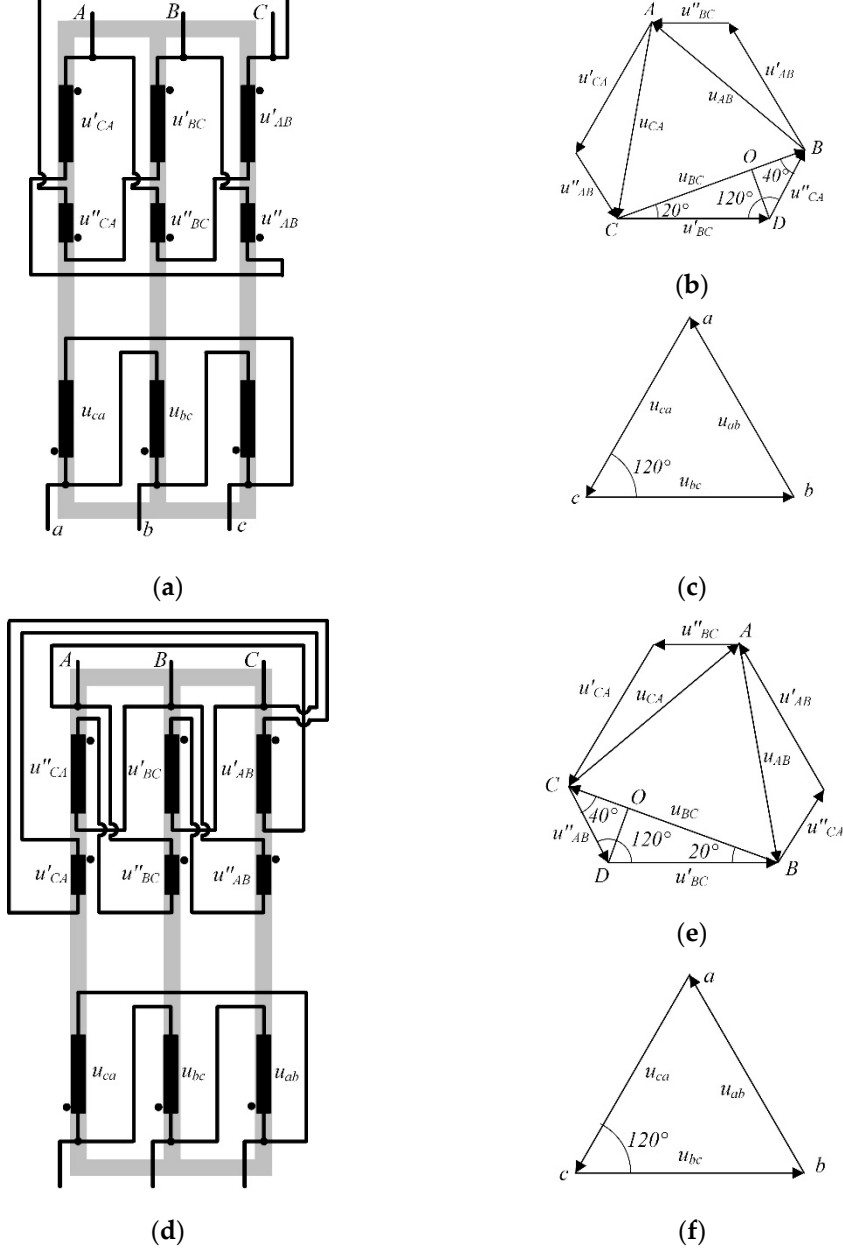

**Figure 2.** Winding connections in phase-shift transformers. (**a**) +20° winding circuit; (**b**) +20° primary voltage vector diagram; (**c**) +20° secondary voltage vector diagram; (**d**) −20° winding circuit; (**e**) −20° primary voltage vector diagram; (**f**) −20° secondary voltage vector diagram.

In a +20° vector shift, ratios of winding parts can be found as follows [12]: vector length $U'_{AB}$, $U'_{BC}$, $U'_{CA}$ equals 1 p.u. The angle between the vectors $U'_{BC}$ and $-U''_{CA}$ equals that between the line *ac* and the vector $U_{BC}$ (Figure 2c), which is 120°. Then, the angle DBO equals 40°. Now let us draw the perpendicular line DO to the segment BC and analyze resultant triangle DOC:

$$DO = DC \cdot \sin(20°) = 1 \cdot 0.342 = 0.342 \text{ p.u.,}$$

$$OC = DC \cdot \cos(20°) = 1 \cdot 0.9397 = 0.9397 \text{ p.u.}$$

For the triangle BDO:

$$BD = DO / \sin(40°) = 0.342 / 0.6458 = 0.5321 \text{ p.u.,}$$

$$BO = DO / tg(40°) = 0.342 / 0.839 = 0.4076 \text{ p.u.,}$$

then

$$BC = OC + BO = 0.9397 + 0.4076 = 1.3473 \text{ p.u.}$$

Thus, the total length of the secondary transformer winding equals

$$1 + 0.5321 = 1.5321 \text{ p.u.}$$

In the case of a $-20°$ vector shift (Figure 2d), the ratio between primary transformer winding parts remains the same; this shift is attained by altering the primary winding connection points.

Thus, the winding ratio is 65% of the turns in the greater part and 35% of the turns in the lesser part.

### 3.3. Three-Level Active Front-End

Active front-ends (AFE) are integral to high-power regenerative electric drives. AFEs have several names commonly used in the literature: PWM boost rectifiers, voltage source rectifiers (VSR), grid inverters/converters, regenerative rectifiers, and bidirectional converters. Researchers and engineers across the world continue to improve AFEs. The motivation here is that for state-of-the-art power converters, AFEs represent the most effective way to manage high power flows. AFEs coupled with multilevel converter topologies and advanced microprocessor-based control systems offer the best quality and efficiency of electric power conversion. Below are the major factors of AFE popularity:

- bidirectional electric power flow control with configurable $\cos(\varphi)$ values;
- compliance with international electromagnetic compatibility standards [13];
- high efficiency rating at 95% to 98% [14].

Figure 3 shows a three-level AFE based on a neutral point clamped (NPC) converter topology [15]. The topology is based on a three-phase bridge that contains 12 fully controllable semiconductor switches $VT_1$–$VT_{12}$, 12 flyback diodes $VD_1$–$VD_{12}$, 6 clamped diodes $VD_{1c}$–$VD_{6c}$ and two full capacitances $C_{dc1}$ and $C_{dc2}$. The total of $u_{dc1}$ and $u_{dc2}$ determines the DC link voltage $u_{dc}$. Only two of the four switches in each bridge leg can be 'on' at a time; they connect the potentials $u_{dc1}$ and $u_{dc2}$ to the load phase. The switching order depends on the modulation algorithm (signals $S_{a1-4}$, $S_{b1-4}$ and $S_{c1-4}$), which further determines the three-phase AFE PWM voltage waveform ($u_a$, $u_b$ and $u_c$). Mathematical description of three-phase AFEs follows well-known methodology and will not be covered herein.

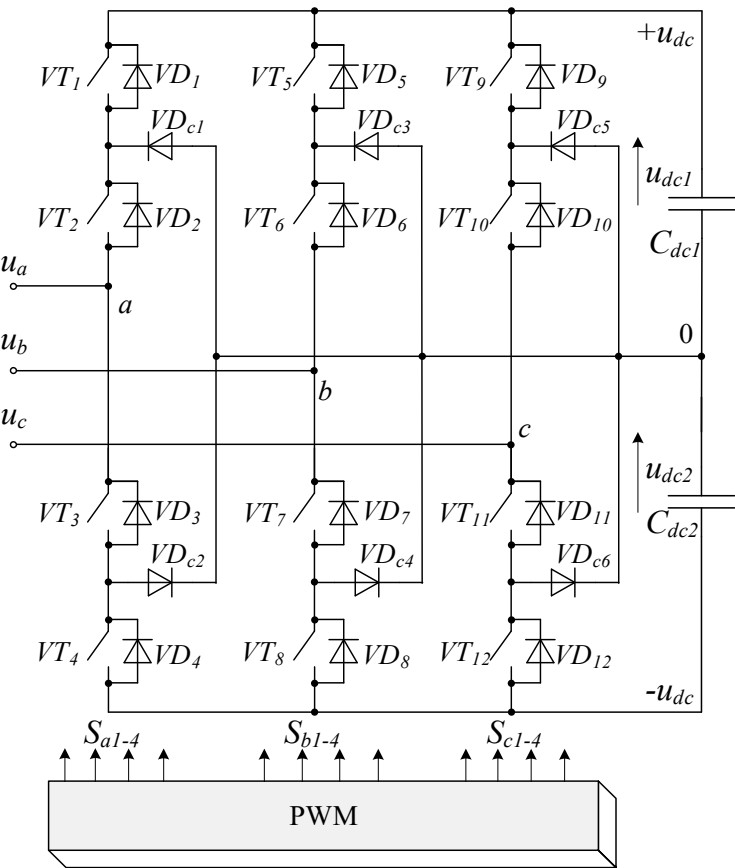

**Figure 3.** Three-level NPC topology.

### 3.4. Programmed Pulse-Width Modulation (PWM)

AFEs use commonly known AC voltage modulation methods, including six-pulse modulation, sinusoidal PWM, space vector modulation, and programmed PWM [16,17]. In case of high power (>1 MW) and medium voltage (3.3 kV to 10 kV), the switching frequency of semiconductor switches in the FC has to be limited to about 450 Hz [18]. In this case, only programmed PWM can modulate an AFE AC voltage waveform to be electromagnetically compatible with the grid [19]. Besides, there has not, so far, been any clear guidelines on how to rate the individual voltage and current harmonic factors after the 50th harmonic, which are important for high-power nonlinear equipment [20].

The object under consideration uses programmed PWM voltage waveform in three three-level AFEs with harmonics 17 and 19 eliminated, which corresponds to an average switching frequency of ~150 Hz. The scientific literature refers to such programmed PWM as selective harmonic elimination pulse-width modulation (SHEPWM) [21].

Figure 4 shows a typical voltage waveform with quarter-wave symmetry in a three-level converter that uses programmed PWM [22–24]. The programmed waveform is attained by switching the semiconductor modules at the right time at the switching angles $\alpha_1, \alpha_2, \ldots, \alpha_N$, whereby N such modules within $[0, \pi/2]$ must be switched over a quarter-cycle provided that $0 < \alpha_1 < \alpha_2 < \ldots < \alpha_N < \pi/2$. N can be found from [25]:

$$N = \frac{f_{swave}}{f}, \tag{1}$$

where $f_{swave}$ is the average semiconductor switching frequency and f is the AFE voltage frequency.

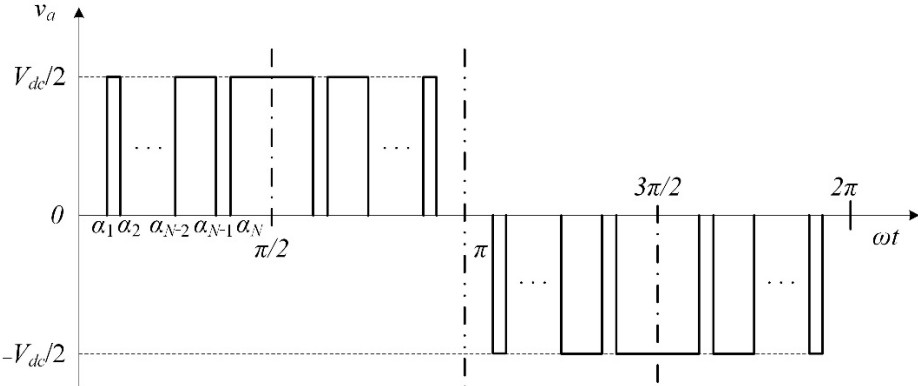

**Figure 4.** Typical programmed PWM waveform of a three-level AFE.

Thus, programmed PWM has a variable period that depends on the time between the preconfigured switching angles [26].

By applying Fourier transform, this signal can be written as

$$V_{ao}(\omega t) = \frac{a_0}{2} + \sum_{n=1}^{\infty} [a_n \cos(n\omega t) + b_n \sin(n\omega t)], \tag{2}$$

where n is the harmonic number and $a_n$ иb_n are Fourier series coefficients.

$$\begin{cases} a_n = \frac{1}{\pi} \int_0^{2\pi} V_{ao}(\omega t) \cos(n\omega t) d(\omega t) & n = 0, 1, \ldots, \infty \\ b_n = \frac{1}{\pi} \int_0^{2\pi} V_{ao}(\omega t) \sin(n\omega t) d(\omega t) & n = 1, 2, \ldots, \infty \end{cases}. \tag{3}$$

Given that the waveform features quarter-wave symmetry, Fourier transform only leaves odd sinusoidal components $b_n$:

$$\begin{cases} a_n = 0 \\ b_n = \begin{cases} 0, & n = \text{even} \\ \frac{4}{n\pi} \frac{V_{dc}}{2} \sum_{k=1}^{n} (-1)^{k+1} \cos(n\alpha_k), & n = \text{odd} \end{cases} \end{cases}. \tag{4}$$

Equation (3) defines the relation between the switching angles and the AC voltage harmonic spectrum of a three-level AFE as

$$\begin{cases} H_1 = \sum_{k=1}^{N} (-1)^k \cdot \cos(\alpha_k) = \frac{\pi}{4} \cdot M \\ \vdots \\ H_n = \sum_{k=1}^{N} (-1)^k \cdot \cos(n \cdot \alpha_k) = 0 \end{cases}, \tag{5}$$

where $H_n$ is the level of the nth harmonic, M is the modulation index and $\alpha_k$ is the switching angle number ranging from 1 to n.

From the equation system (4), the number of eliminated harmonics can be found by the formula

$$n = N - 1. \tag{6}$$

The modulation index M ranges from 0 to $M_{max}$, which can be found by the formula

$$M_{max} = b_1/(V_{dc}/2) = \frac{4}{\pi}. \tag{7}$$

In order to solve this nonlinear algebraic equation to obtain the unknown variable, many of the methods used in optimization solvers are proposed [27–30]. For the sake of

generality and easy implementation for general microcomputers, a trust-region dogleg algorithm is employed in order to solve the SHEPWM equations [31,32].

By default, the fsolve function in Matlab chooses the trust-region dogleg algorithm, and, therefore, can be applied in the m-file codes with the objective function (5) to solve the nonlinear equations of SHEPWM. For each $M_a$, this method runs in increments of 0.01 from 0 to 1. After a suitable initial value is given, one switching pattern can be achieved.

The initial values for $N$ switching angles (here, $N$ = odd), unipolar SHEPWM can be derived as

$$\begin{cases} \alpha^0_{2k-1} = 30° + 120° \cdot k/(N+1) - \Delta\alpha \\ \alpha^0_{2k} = 30° + 120° \cdot k/(N+1) + \Delta\alpha \\ \alpha^0_N = 90° - \Delta\alpha \end{cases} \tag{8}$$

where $k = 1, 2, \ldots, (N-1)/2$ and $\Delta\alpha$ is an optimal small angle, which is used to avoid singularity of the fsolve function. After testing, $\Delta\alpha = 0.3 \sim 0.5$ and some other values can ensure iterative convergence.

The equation system (5) produces four patterns for harmonic elimination:

- 5, 7, 17, and 19 in Pattern 1 (250 Hz);
- 17 and 19 in Pattern 2 (150 Hz);
- 17, 19, 35, and 37 in Pattern 3 (250 Hz);
- 5, 7, 17, 19, 35, and 37 in Pattern 4 (350 Hz).

Figure 5 visualizes the results.

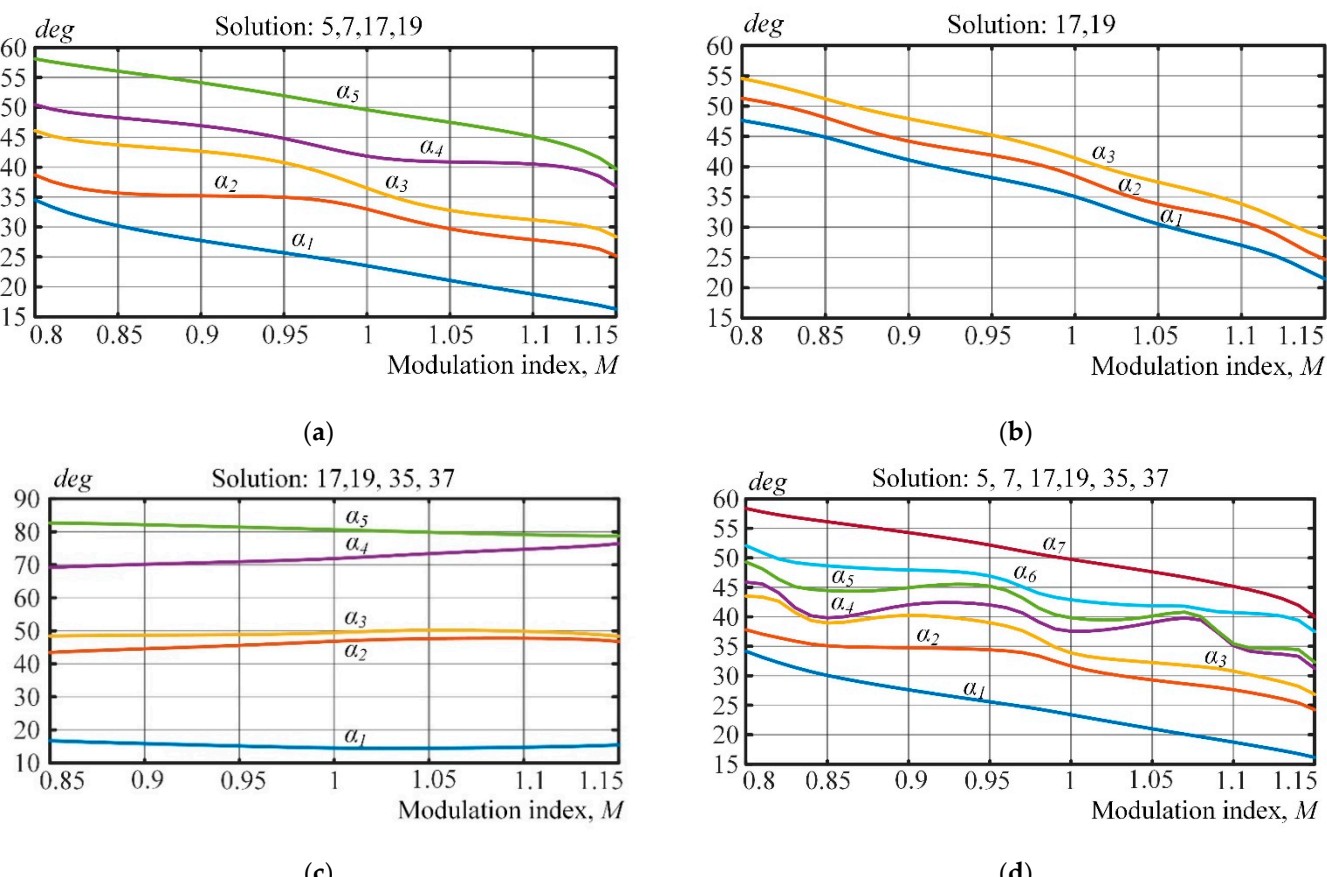

**Figure 5.** Calculations of four AFE switching patterns. (**a**) Pattern 1 with harmonics 5, 7, 17 and 19 eliminated. (**b**) Pattern 2 with harmonics 17 and 19 eliminated. (**c**) Pattern 3 with harmonics 17, 19, 35 and 37 eliminated. (**d**) Pattern 4 with harmonics 5, 7, 17, 19, 35 and 37 eliminated.

## 4. Control System

Most AFE control systems are based on the so-called voltage-oriented control (VOC). Linear controllers are either proportional (P) or proportional-integral (PI).

Figure 6 shows the circuit diagram of the control system used in the analyzed object. One peculiar feature of this circuit is that the three-phase current signals $i_{abc}$ and voltage signals $u_{abc}$ are measured at the primary side of the $T^{0°}$, $T^{20°}$, $T^{-20°}$ phase-shift transformers. This is possible thanks to the absence of other power-consuming equipment. Figure 6 uses the following notation: $T^{0°}$, $T^{20°}$, $T^{-20°}$ are phase-shift transformers; $PLL^{0°}$, $PLL^{20°}$, $PLL^{-20°}$ are units that synchronize voltages to the secondary windings of phase-shift transformers; $i_{abc}$ and $u_{abc}$ are the measured instantaneous phase currents and values on the primary side of the phase-shift transformers in *abc* coordinates; $\theta$ is the grid voltage space vector angle; $\theta^{0°}$, $\theta^{20°}$, $\theta^{-20°}$ are the calculated voltage space vectors for the secondary windings of phase-shift transformers; $i_{dq}$ are measured instantaneous phase currents and values on the primary side of the phase-shift transformers in *dq0* coordinates; $i_{dq}{}^{0°}$, $i_{dq}{}^{20°}$, $i_{dq}{}^{-20°}$ are the measured instantaneous phase currents of AFEs in *dq0* coordinates; $i_{dq\mathrm{ref}}{}^{0°}$, $i_{dq\mathrm{ref}}{}^{20°}$, $i_{dq\mathrm{ref}}{}^{-20°}$ are the configured phase currents of AFEs in *dq0* coordinates; $u_{dq}{}^{0°}$, $u_{dq}{}^{20°}$, $u_{dq}{}^{-20°}$ are the measured instantaneous phase voltages of AFEs in *dq0* coordinates; $u_{dc}{}^{0°}$, $u_{dc}{}^{20°}$, $u_{dc}{}^{-20°}$ are the measured instantaneous voltages of DC link capacitors in AFEs; $u_{dc\mathrm{ref}}{}^{0°}$, $u_{dc\mathrm{ref}}{}^{20°}$, $u_{dc\mathrm{ref}}{}^{-20°}$ are the configured AFE DC link capacitor voltages; $m^{0°}$, $m^{20°}$, $m^{-20°}$ are the AFE modulation indices; $\alpha^{0°}$, $\alpha^{20°}$, $\alpha^{-20°}$ are the phase shifts between secondary windings of phase-shift transfers and phase voltages of AFEs; $L_{AFE}$ is the AFE input inductance; $L_{load}$ is the FC input inductance; LPF are lowpass filters.

This paper does not dwell upon the synthesis of a current and voltage control system. Controllers are synthesized by cascade control.

Table 4 shows the parameters of the DC link voltage PI controller $K_{pdc}$ and $K_{idc}$, as well as the parameters of the current vector PI controller $K_{pi}$ and $K_{ii}$ in the orthogonal *dq0* axes.

**Table 4.** Controller parameters.

| $\mathbf{K_{pdc}}$ | $\mathbf{K_{idc}}$ | $\mathbf{K_{pi}}$ | $\mathbf{K_{ii}}$ |
|---|---|---|---|
| 0.5 | 20 | 1.5 | 25 |

Voltage and current harmonic distortions were analyzed in Matlab/Simulink. The developed Matlab/Simulink model followed the description above and is not detailed here.

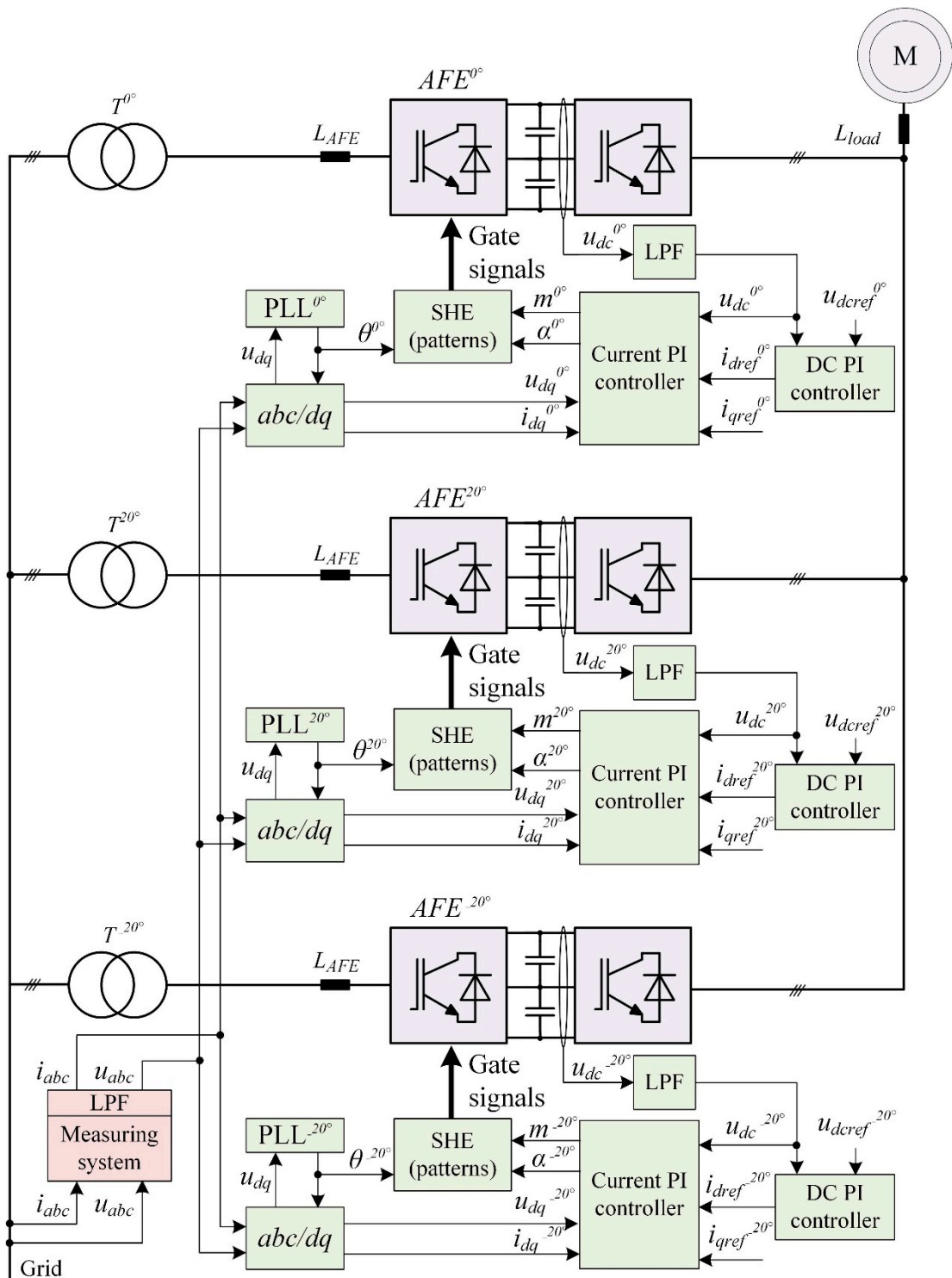

**Figure 6.** Object control system.

## 5. Simulation Results and Discussion

Voltage and current harmonic distortions were analyzed in Matlab/Simulink. The developed Matlab/Simulink model followed the description above and is not detailed herein for complexity and space considerations. Figures 7–9 show the most visual results of simulating the currents of the object with four patterns obtained in Section 3.

Figure 7 shows the results of simulating the three-level AFE-consumed rated current $i_{aAFE}$ in the secondary winding of one of the three phase-shift transformers. Analysis of

the data produced calculations of the total harmonic distortions (THD) and individual harmonic factors for up to the 50th $i_{aAFE}$ harmonic.

Figure 7a,c clearly shows $i_{aAFE}$ to have ~30% THD in Patterns 1 and 4, which is about 1.5 times less than in Patterns 2 and 3. This is because that the individual harmonic factors of $i_{aAFE}$ harmonics 5 and 7 have the greatest effect in the visible spectrum. Apparently, eliminating harmonics 35 and 37 (Patterns 3 and 4) results in no significant improvement in the harmonic spectrum of $i_{aAFE}$. The reason is that the inductive reactance of the AFE input is sufficient to passively filter out the harmonics on this order.

Simulation results were performed for a rate modulation index 1.107. Figures 7–9 show the obtained current and voltage THD considering harmonics up to 50th order.

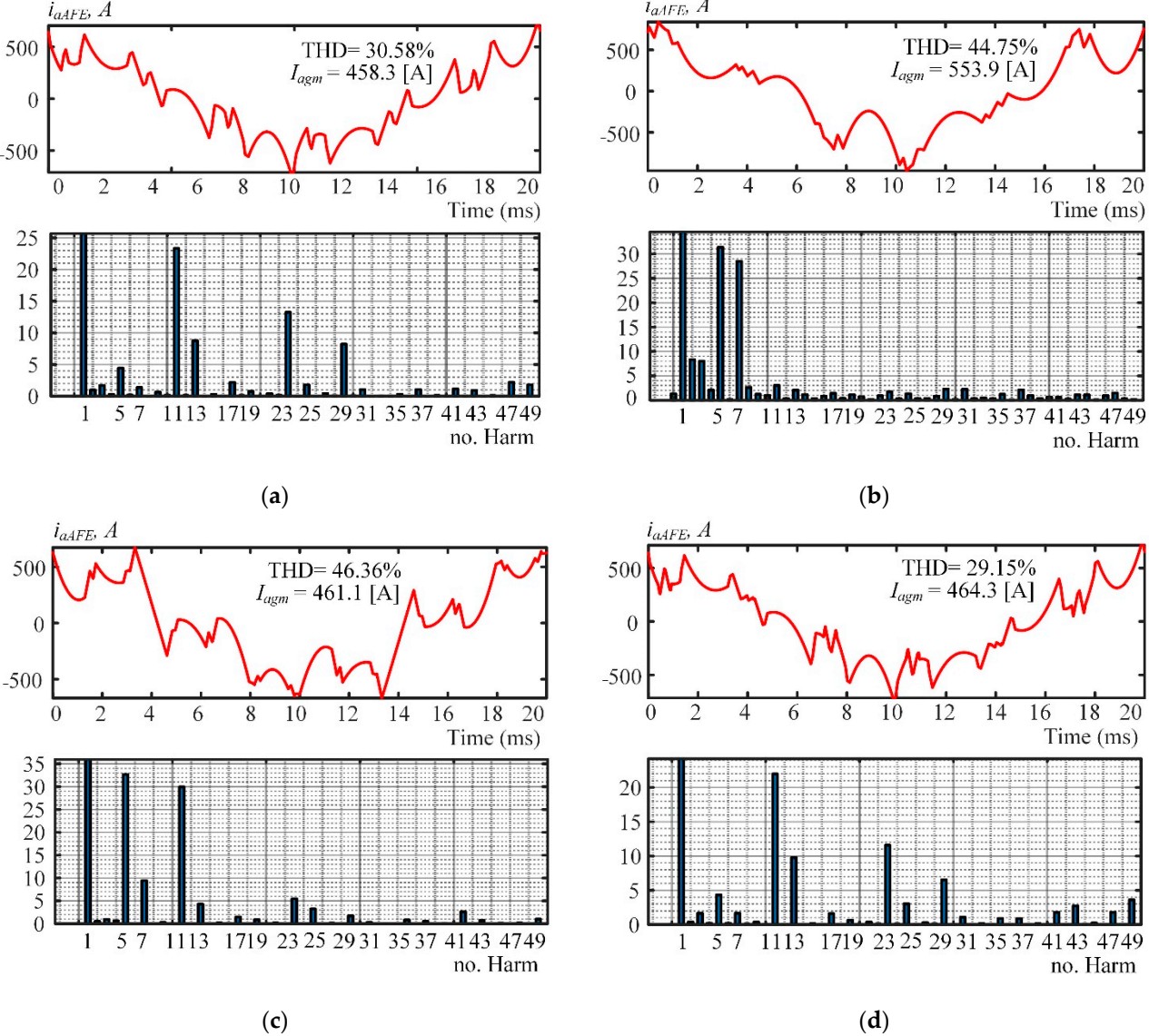

**Figure 7.** Oscillograms of rated AFE current $i_{aAFE}$ in the secondary winding of a single phase-shift transformer. (**a**) Pattern 1; (**b**) Pattern 2; (**c**) Pattern 3; (**d**) Pattern 4.

Therefore, Pattern 1 with harmonics 5, 7, 17 and 19 eliminated is recommended to prevent the secondary windings of the phase-shift transformers from overheating. Pattern 2, which is currently in use and eliminates harmonics 17 and 19, is not recommended.

Figure 8 shows the results of simulating the rated current $i_{ag}$ drawn from the grid by the three three-level AFEs in an 18-pulse connection. Figure 8c clearly shows that the THD of $i_{ag}$ has the best value (3.66%) in Pattern 3, which is 1.39% less than in the case of the

recommended Pattern 1 (see Figure 8a). The reason is that 18-pulse connection filters out harmonics of secondary-winding currents except $18n \pm 1$ (n = 1, 2, ... , ∞).

Thus, the quality of the current $i_{ag}$ in the recommended Pattern 1 (harmonics 5, 7, 17, and 19 eliminated) is not far below that in Pattern 3. Compared to Patterns 2 (Figure 8b) and 4 (Figure 8d), the quality of $i_{ag}$ in Pattern 1 has better THDs and individual harmonic factors for up to the 50th harmonic.

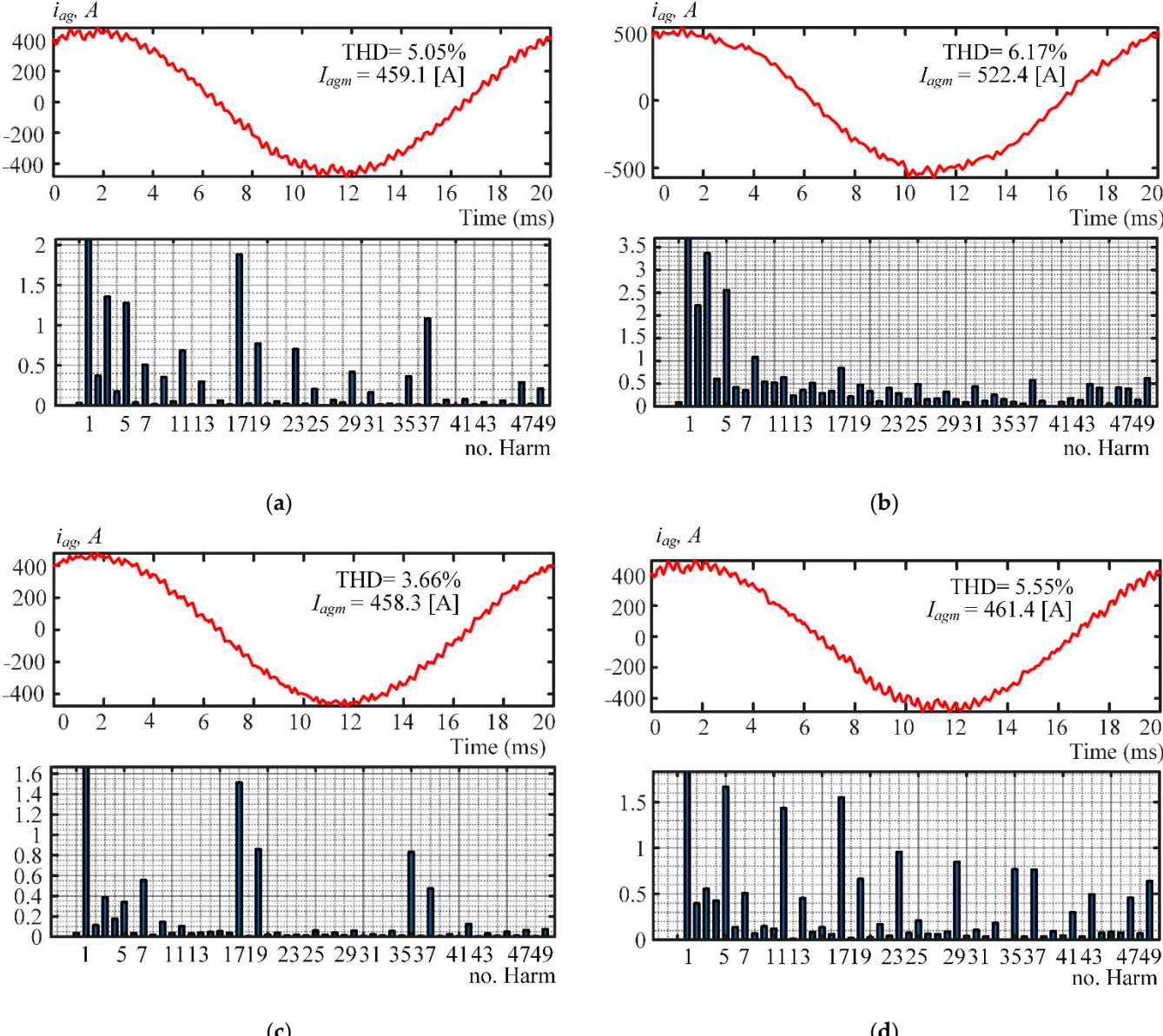

**Figure 8.** Oscillograms of the rated phase current $i_{ag}$ drawn from the grid by the three three-level AFEs. (**a**) Pattern 1; (**b**) Pattern 2; (**c**) Pattern 3; (**d**) Pattern 4.

Figure 9 shows the readings of phase-to-phase AFE voltage in the secondary winding of the −20° phase-shift transformer. Figure 9a demonstrates that the THD of $u_{abAFE}$ has a subpar value (29.70%) in Pattern 1, and an increase of 3–5% from Patterns 2, 3 and 4 (see Figure 8b,c). However, this does not cause significant issues with the AFE current as shown in Figures 7 and 8.

As shown in Figures 7–9, the four patterns produced very similar results, but in Pattern 1, the switching angle provided optimal current harmonic distortions in the 18-pulse connection. On the other hand, Pattern 3 of the SHEPWM strategy provided the best

current in the grid. It could be successfully used in AFE operations while the phase-shifting transformers are running at nominal temperatures.

In the spectra under consideration, one can note the harmonics that the 18-pulse connection should have filtered out (see harmonic spectra in Figures 7–9). Their presence could be due to the imperfect synchronization of three-level AFE control systems and errors in finding the phase-shift transformer winding ratios, which resulted in inaccurate semiconductor module switching angles.

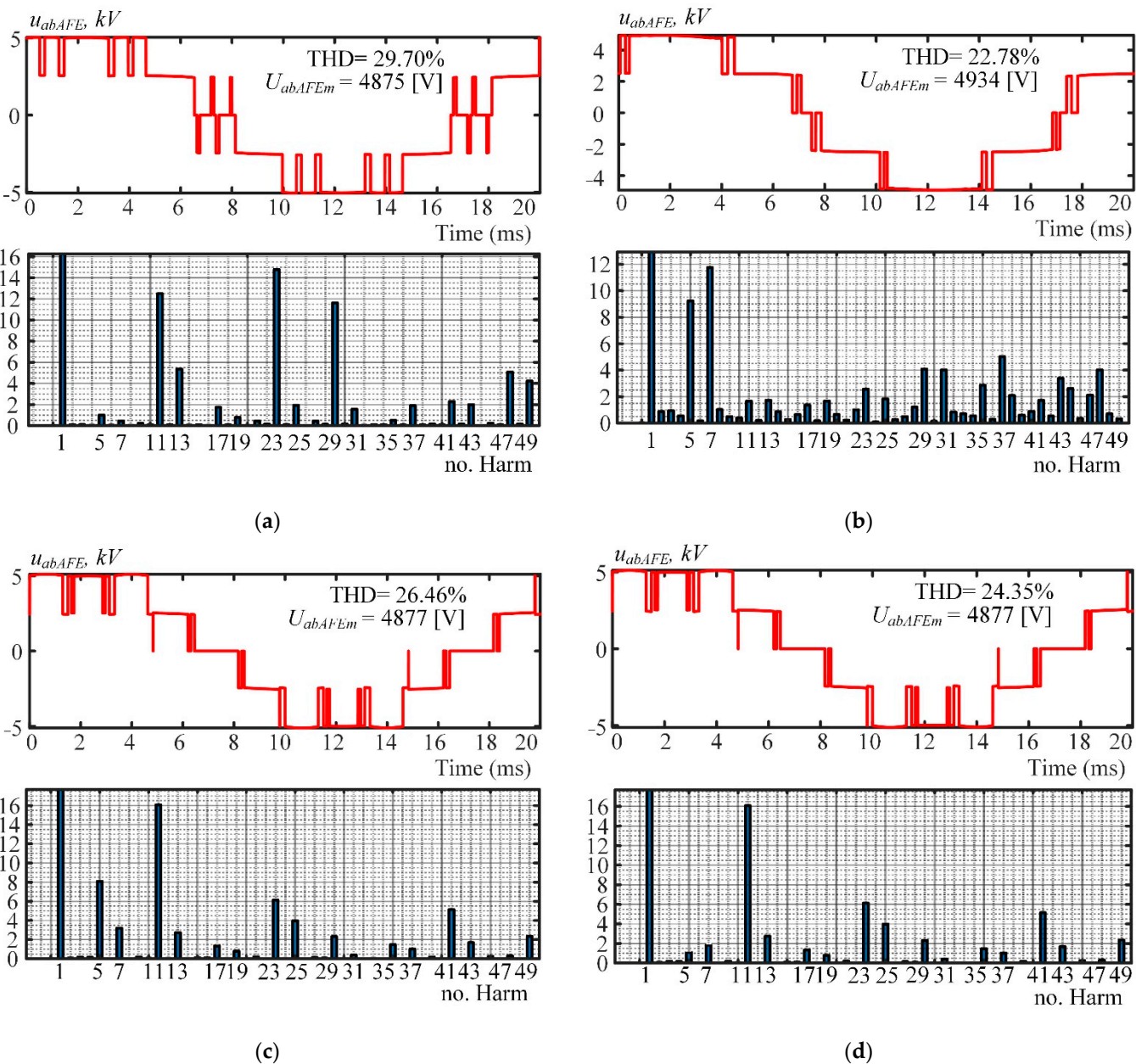

**Figure 9.** Oscillograms of the line AFE voltage $u_{abAFE}$. (**a**) Pattern 1; (**b**) Pattern 2; (**c**) Pattern 3; (**d**) Pattern 4.

## 6. Conclusions

Low switching selective harmonic elimination by programmed PWM for multipulse connections of AFEs can produce a high-quality current harmonic spectrum. The proposed implementation of the SHEPWM method can be applied to NPC topology of AFEs. A trust-region dogleg algorithm was applied to calculate switching patterns of SHEPWM technique and one universal formula for calculating initial values of SHEPWM technique

is proposed. Several results of 18-pulse AFE rectifiers proved the validity of the algorithm with this formula.

To the authors' knowledge, this paper is the first to analyze current harmonic distortions in an 18-pulse connection of three-level AFEs featuring a programmed PWM with four patterns that eliminate the following harmonics: 5, 7, 17 and 19; 17 and 19; 17, 19, 35 and 37; and 5, 7, 17, 19, 35 and 37. The results helped find the best such waveform to prevent phase-shift transformers from overheating, i.e., one with harmonics 5, 7, 17 and 19 eliminated. These results are highly versatile and could be used in similar research of other multipulse connections in order to ensure the electromagnetic compatibility of AFEs with programmed PWM.

**Author Contributions:** Methodology and validation, A.S.M.; writing—original draft preparation, A.S.M. and T.J.; writing—review and editing, A.S.M. and T.J.; investigation, A.S.M. and T.J.; project administration and supervision, A.A.R. and V.R.G.; supervision, A.S.M.; software, T.A.L. All authors have read and agreed to the published version of the manuscript.

**Funding:** The research was funded by RFBR and Chelyabinsk Region, project number 20-48-740008.

**Conflicts of Interest:** The authors declare no conflict of interest.

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
