# Peer review of "Finding the Best Programmable PWM Pattern for Three-Level Active Front-Ends at 18-Pulse Connection"

_machines, doi:10.3390/machines9070127_

Round 1

Reviewer 1 Report

The main idea of present solution is for me unclear. I think it should be better explained.

In my opinion, the presented algorithm should be confirmed by experimental research. However, it is not clear to me whether the authors have conducted such research. In line 265 I found sentence “Experimental results were performed for a rate modulation index 1.107. Figures 7-9 shows the obtained current and voltage THD considering harmonics up to 50th order”.

Sentence in Conclusions (line 304) in my opinion looks strange “Selective harmonic elimination programmed PWM method for multi-pulse connections of AFEs is determined to achieve high-quality current harmonic spectrum with low switching.”

I think that after experimental verification it will be a good material for publication.

Author Response

The authors agree with the comments and thank the Reviewer for a close reading of the manuscript.

Moderate English changes has been done.

1. The main idea of present solution is for me unclear. I think it should be better explained.

Title of the paper has been changed to better represent the contributions of the paper; it now reads, Finding the Best Programmable PWM pattern for Three-Level Active Front-Ends at 18-Pulse Connection

2. In my opinion, the presented algorithm should be confirmed by experimental research. However, it is not clear to me whether the authors have conducted such research. In line 265 I found sentence “Experimental results were performed for a rate modulation index 1.107. Figures 7-9 shows the obtained current and voltage THD considering harmonics up to 50th order”. I think that after experimental verification it will be a good material for publication.

Experimental results have been obtained by mathematical modeling at a rate modulation index of 1.107 (Figures 7-9). The results demonstrated good performance of the programmed PWM pattern with the elimination of the 5, 7, 17, and 19 harmonics. The pattern was recommended for the industrial object to prevent the phase-shift transformers from overheating. After the implementation of this solution, a decrease in overheating was noted.

The adequacy of scientific conclusions and recommendations can be proved by a scientifically statement and mathematical modeling. and it is confirmed by the positive effect after implementation on the object. Research performed directly on the real object and operating equipment has been classified by the Company and is therefore not publicly available.

3. Sentence in Conclusions (line 304) in my opinion looks strange “Selective harmonic elimination programmed PWM method for multi-pulse connections of AFEs is determined to achieve high-quality current harmonic spectrum with low switching.”

The sentence has been corrected below:

«Low switching selective harmonic elimination by programmed PWM for multi-pulse connections of AFEs can produce a high-quality current harmonic spectrum» 

Reviewer 2 Report

This paper presents the current harmonics distortion in 18-pulse connection of three level AFE with programmable PWM. This paper is organised well however, i have following concerns

  1. The contribution of paper is still not clear as authors have mentioned many things in introduction and abstract.
  2. The contribution of paper must match with the tittle of the paper as reader can easily understand. 
  3. "Compared to connections that use six 37 pulses, multi-pulse connections have two core advantages"  Authors must cite a paper with each advantage. 
  4. Also cite a paper from where the points were referenced about capability of three level inverter. 
  5. "Literature review suggests the following technical solutions are relevant today [6-10]:" Authors are suggested to add following reference in this sentence. Additionally, authors are also suggested to add reference seperately with each sentence instead of writing in collective manner as it will increase readability of article. 

"Resonance damping for an LCL filter type grid-connected inverter with active disturbance rejection control under grid impedance uncertainty" DOI: https://doi.org/10.1016/j.ijepes.2019.02.004.

6. Your contribution is the selection of best programmable PWM configuration to overcome the phase shift transformer from overheating. If this is so then you may need to change the title of paper. 

7. Please cite reference paper with each sentence of introduction where required. Most of the introduction is not cited properly. 

Author Response

The authors agree with the comments and thank the Reviewer for a close reading of the manuscript.

Moderate English changes have been done.

1. The contribution of paper is still not clear as authors have mentioned many things in introduction and abstract;

2. The contribution of paper must match with the tittle of the paper as reader can easily understand;

6. Your contribution is the selection of best programmable PWM configuration to overcome the phase shift transformer from overheating. If this is so then you may need to change the title of paper.

Title of the paper has been changed to better represent the contributions of the paper; it now reads, Finding the Best Programmable PWM pattern for Three-Level Active Front-Ends at 18-Pulse Connection

Less overheating is one of the possible positive outcomes of applying the best PWM pattern, which was based on current harmonic distortions analysis. In our opinion, a title focused on transformers may cause confusion as transformers are not the core topic: the paper is more about AFE, PWM, power quality and 18-pulse connection.

3. "Compared to connections that use six 37 pulses, multi-pulse connections have two core advantages"  Authors must cite a paper with each advantage. 

The references [4] and [5] have been separated.

4. Also cite a paper from where the points were referenced about capability of three level inverter

The reference below has been added:

Nabae, A.; Takahashi I.; Akagi H. New neutral-point-clamped PWM inverter. IEEE Transactions on Industrial Applications 1981, IA–17, 5, pp. 518–523.

5. "Literature review suggests the following technical solutions are relevant today [6-10]:" Authors are suggested to add following reference in this sentence. Additionally, authors are also suggested to add reference seperately with each sentence instead of writing in collective manner as it will increase readability of article. 

"Resonance damping for an LCL filter type grid-connected inverter with active disturbance rejection control under grid impedance uncertainty" DOI: https://doi.org/10.1016/j.ijepes.2019.02.004

The references [6-10] have been separated.

The reference below has been added:

[10] Muhammad Saleem; Ki-Young Choi; Rae-Young Kim. Resonance damping for an LCL filter type grid-connected inverter with active disturbance rejection control under grid impedance uncertainty. International Journal of Electrical Power & Energy Systems 2019, 109, pp. 444–454.

7. Please cite reference paper with each sentence of introduction where required. Most of the introduction is not cited properly. 

The placement of references in the introduction has been changed.

Round 2

Reviewer 1 Report

In my opinion, experimental research is research of a physical object and not simulation research. Therefore, in order not to mislead the reader, the name simulation test should be used.

Author Response

The authors agree with the comments and the manuscript name was modified "Simulation Research to Finding the Best Programmable PWM Pattern for Three-Level Active Front-Ends at 18-Pulse Connection".

Reviewer 2 Report

The paper has been improved considerably. All the comments are addressed properly. Thank you. 

Author Response

The authors thank the Reviewer for a close reading of the manuscript.